# Development of an Enzyme-Linked Immunosorbent Assay (ELISA) for Accurate and Prompt Coronavirus Disease 2019 (COVID-19) Diagnosis Using the Rational Selection of Serological Biomarkers

**DOI:** 10.3390/diagnostics11111970

**Published:** 2021-10-23

**Authors:** Theano Lagousi, John Routsias, Vana Spoulou

**Affiliations:** 1First Department of Paediatrics, “Aghia Sophia” Children’s Hospital, Immunology and Vaccinology Research Laboratory and Infectious Diseases Department “MAKKA”, Athens Medical School, 11527 Athens, Greece; vspoulou@med.uoa.gr; 2University Research Institute for the Study of Genetic & Malignant Disorders in Childhood, First Department of Paediatrics, Aghia Sofia Children’s Hospital, Athens Medical School, 11527 Athens, Greece; 3Department of Microbiology, Athens Medical School, 11527 Athens, Greece; jroutsias@med.uoa.gr

**Keywords:** SARS-CoV-2, antibodies, serological tests

## Abstract

Prompt COVID-19 diagnosis is urgently required to support infection control measures. Currently available serological tests for measuring SARS-CoV-2 antibodies use different target antigens, although their sensitivity and specificity presents a challenge. We aimed to develop an “in-house” serological ELISA to measure antibodies against SARS-CoV-2 by combining different protein antigens. Sera (*n* = 44) from COVID-19-confirmed patients were evaluated against different SARS-CoV-2 protein antigens and all potential combinations using ELISA. Patients’ sera were also evaluated against commercially available ELISA diagnostic kits. The mixture containing RBD 2.5 μg/mL, S2 1 μg/mL and N 1.5 μg/mL was found to be the most potent. Plates were incubated with patients’ sera (1:100), and goat anti-human alkaline phosphatase-conjugated IgG, ΙgM and IgA antibody was added. The cut-off value for each assay was determined using the mean optical density plus two standard deviations of pre-pandemic controls. The “in-house” ELISA displayed 91% sensitivity and 97% specificity for IgG antibodies, whereas its sensitivity and specificity for IgM and IgA were 75% and 95% and 73% and 91%, respectively. The “in-house” ELISA developed here combined three SARS-CoV-2 antigens (RBD, S2 and N) as capture antigens and displayed comparable and even higher sensitivity and specificity than otherwise quite reliable commercially available ELISA diagnostic kits.

## 1. Introduction

Coronavirus disease 2019 (COVID-19) is a significant infectious disease of respiratory distress declared as a pandemic, caused by the severe acute respiratory syndrome coronavirus 2 (SARS-CoV-2) [1,2,3]. It originated in December 2019 in Wuhan, Hubei Province, China, where the first COVID-19 cases were identified [4]. On September 15th, according to the World Health Organization’s official daily COVID-19 Situation Reports, there were over 225 million confirmed cases and over 4.5 million deaths reported worldwide [5]. Most people infected with the virus experience mild to moderate respiratory illness and recover without requiring special treatment. However, some will become seriously ill with or without pneumonia, fever, cough and respiratory distress and require medical care. Older people and individuals of all ages with underlying chronic medical conditions are most likely to present with more severe disease [6].

Symptomatic, asymptomatic and pre-symptomatic individuals may spread the virus [7]. Considering that the vast majority of people with COVID-19 have mild or asymptomatic disease, asymptomatic transmissions are responsible for almost half of all COVID-19 cases [8,9]. Symptoms of SARS-CoV-2 infection regularly appear 2–14 days post-exposure [10], albeit the incubation period varies significantly among different patients’ populations; thus, viral transmission may be undetectable.

Early and accurate diagnostic testing for COVID-19 is critical for tracking SARS-CoV-2 spread, recording the epidemiology of the virus, the management of each case, controlling the transmission and implementing quarantine rules.

Nucleic-acid-based testing technologies that use real-time polymerase chain reaction (RT-PCR) to detect SARS-CoV-2 remain the gold standard, as highly sensitive and accurate diagnostic assays to confirm COVID-19 in the early phase of disease [11,12]. RT-PCR has identified the SARS-CoV-2 RNA genome in throat and nasal specimens of infected individuals [13]. Notably, the amount of virus detected in the upper respiratory tract secretions of COVID-19 patients peaks 7 days post-symptom onset, but may drop significantly below the threshold of RT-PCR detection later in the course of the disease [8,10,11,12]. In addition, false negative results cannot be excluded when missing the time-window of viral replication.

Thus, supplementary screening assay techniques for the detection of SARS-CoV-2, despite its low viral load, are urgently required to guarantee the early diagnosis of all COVID-19 cases. The assessment of specific antibodies against SARS-CoV-2, including both immunoglobulin G (IgG) and M (IgM), as well as A (IgA), which are induced quickly post-disease onset, offers an alternate highly sensitive and accurate diagnostic tool that may offset the limitations of RT-PCR [14]. Serological assays may enable serological surveillance, detect those who have previously been infected, assess natural acquired immunity and record the extent of COVID-19 spread. Furthermore, serological assays may facilitate contact tracing, evaluate vaccine-induced immunity, detect patients that are admitted late from disease onset and identify appropriate convalescent plasma donors.

Serological assays for the diagnosis of SARS-CoV-2, in contrast to molecular assays, i.e., RT-PCR, are often less time-consuming, cheaper and are easier to perform by staff without significant laboratory qualifications. Detection of the production of antibodies can be a tool, which either alone or in conjunction with PCR may enhance detection sensitivity and accuracy.

SARS-CoV-2 is an enveloped, non-segmented, single-stranded positive-sense RNA virus that belongs to the Coronaviridae family of the betacoronavirus genus. The SARS-CoV-2 genome has approximately 30 kilobases, encoding for several structural proteins including the spike (S), the envelope (E), the membrane (M), the nucleocapsid (N) and non-structural proteins [15,16]. The S protein comprises two functional domains that are involved in cell attachment and fusion of the viral and cellular membrane, S1 and S2 subunit. The spike proteins of SARS-CoV-2 commonly bind to the human angiotensin-converting enzyme 2 (ACE2) protein as a host receptor through their receptor-binding domain (RBD) in SARS-CoV-2 S protein. The binding of the S protein with the host–receptor promotes permanent conformational changes in its structure, enabling membrane fusion [15,16]. The N structural protein attaches to the viral RNA genome, generating a capsid. The M protein is the most prevalent structural protein of the virus and forms its distinct shape and structure. The E antigen is involved in new virion assembly and release and promotes virus pathogenesis [17,18,19].

Several commercial serological kits for measuring SARS-CoV-2 IgG, IgA and IgM antibodies have been approved by the FDA [20]. These assays target immunogenic coronavirus proteins, mainly the N protein, the S protein and the S1 fragment, which may differ significantly regarding its reactogenicity from the S protein as a whole, or the receptor-binding domain (RBD). However, the sensitivity and specificity of enzyme-linked immunosorbent assays (ELISAs) largely rely on the type of viral protein used as a capture antigen. Nevertheless, antibody seroconversion differs in COVID-19 patients based on individual diversity (i.e., disease severity and immune status) and may occur quite late after disease onset. Most COVID-19 patients elicit specific antibodies against SARS-CoV-2 7–11 days after exposure, although significant discrepancies have been recorded [20].

In this study, we aimed to develop a serological approach of high sensitivity and specificity to promptly define a true positive SARS-CoV-2 infection by combining different protein antigens as capture antigens, considering that kinetics of antibodies against distinct protein antigens are not yet fully defined. Secondly, the ELISA developed “in-house” was compared with currently available commercial serological tests for SARS-CoV-2 diagnosis.

## 2. Study Design

### 2.1. Study Population

Sera (N = 50) from patients (aged 18–55) with confirmed COVID-19 were used. COVID-19 had been confirmed by PCR assays, in nasopharyngeal and/or oropharyngeal swab samples. All of the blood samples were collected between 1 and 14 days of disease onset. Serum samples from healthy aged-matched blood donors (N = 150) were used as controls, collected before the outbreak of the pandemic (during the summer period of 2018). A written informed consent was previously obtained from all patients and controls. The Ethics Committee of Aghia Sofia Children’s Hospital approved the study protocol (4578/08-03-21). Sera from children aged 0–14 years with PCR-confirmed COVID-19 were also used (*n* = 15, obtained 8–14 days from disease onset, and n = 7, obtained 1–5 days from disease onset). Sera from 30 aged-matched children with no history of acute infection or underlying disease were used as controls. Written informed consent was obtained from the guardians of all subjects.

### 2.2. Proteins

SARS-CoV-2 antigens were obtained commercially. These included 7 SARS-CoV-2 proteins: Spike protein S1 + S2 (13–686 aa) (purity >90% by SDS-PAGE) (Sino Biological), Spike protein S1 (13–685 aa) (purity >90% by SDS-PAGE) (Abclonal), Spike protein S2 (686–1273 aa) (purity > 90% by SDS-PAGE) (Sino Biological), Spike receptor-binding domain (RBD) 334–527 aa (purity > 95% by SDS-PAGE) (Abclonal), Membrane protein (1–222 aa) (purity >90% by SDS-PAGE) (Abcam), Envelope small membrane protein (1–75 aa) (purity > 95% by SDS-PAGE) (Abclonal), Nucleoprotein (Nucleocapsid protein) (full length 1–419 aa) (purity > 95% by SDS-PAGE) (Abclonal).

### 2.3. Selection of the Most Immunoreactive Protein Antigens

Nighty-six-well plates (Nunc Maxisorp, Rochester, NY, USA) were coated with different protein antigens suspended in phosphate-buffered saline (PBS). The plates were then blocked with 200 µL/well of PBS containing 2% bovine serum albumin (BSA) at 37 °C for 30 min. The blocking solution was removed, and diluted serum samples (1:100 in 2% BSA PBS) were added to the plates for 1 h at 37 °C. Each serum was evaluated against BSA (0.01% PBS) to eliminate non-specific binding. The plates were washed three times with PBS/0.05% Tween 20. Alkaline phosphatase-conjugated goat anti-human IgG (Jackson ImmunoResearch Laboratories, 1:3000) antibody diluted in PBS containing 2% BSA and 0.05% Tween 20 was used to reveal specific human antibodies (IgGs). Preliminary experiments were performed to determine optimal incubation time periods, whereas the concentration of the coated antigens and plasma dilutions for this ELISA were optimized using chessboard titration tests. Antibody binding was assessed with the substrate 4-nitrophenyl-phosphate-disodium salt hexahydrate (Sigma Chemicals, St. Louis, MO, USA) at 405 nm (Chromate reader, Awareness Technology). The cut-off value was determined as the mean plus 2 standard deviations (SDs) of the pre-COVID-19 controls (n = 150). The sample was defined as ELISA-antibody-positive if the OD405 value was 2 SDs above the mean of the controls.

### 2.4. Selection of the Most Immunoreactive Combination of Protein Antigens

Subsequently, various mixtures of all potential combinations of protein antigens were evaluated in terms of their antigenicity among patients’ and controls’ sera. The combination of protein antigens with the highest immunoreactivity was selected for further evaluation. Antibody binding was detected as previously described using substrate 4-nitrophenyl-phosphate-disodium salt hexahydrate (Sigma Chemicals) at 405 nm (Chromate reader, Awareness Technology). The cut-off was determined as the mean plus 2 SDs of the pre-COVID-19 control population. Preliminary results were analyzed to determine the optimal capture antigen concentration and serum dilution.

### 2.5. IgA and IgM Assessment

To assess the ability of the selected assay to detect the distribution of the different antibody isotypes in our study population, different secondary goat anti-human IgA antibodies (Jackson ImmunoResearch Laboratories, 1:1500) and goat anti-human IgM (Jackson ImmunoResearch Laboratories, 1:3000) were used in the optimized SARS-CoV-2 ELISA, as previously described.

### 2.6. Determining Inter-Assay Variability and Repeatability

Subsequently, the inter-assay repeatability of the developed ELISA was determined, using pre-COVID-19 controls (n = 4) and COVID-19 patients’ sera (n = 4). For this aim, serum samples were tested in four distinct assays using the SARS-CoV-2 ELISA described above. Results are shown as the optical density for each antigen/antibody isotype combination (IgG, IgA and IgM). Results are defined as a ratio of the recorded optical density absorbance to the pre-defined cut-off optical density. Values with a ratio above 1 were considered positive.

### 2.7. Sera Heat Treatment for Viral Inactivation

Patient serum was inactivated to exclude any residual virus in the samples. Then, 10 recovered COVID-19 patients and 5 pre-COVID-19 controls were assessed using heat treatment [21,22]. Sera (0.5 mL) were incubated at 56 °C for 30 min while rotationally shaking and centrifuging at 14,000× *g* for 10 min [21]. The supernatant was collected and applied to the SARS-CoV-2 ELISA, as previously described. Both non-treated and treated patient samples were tested. However, virus viability was not further evaluated.

### 2.8. Inhibition of IgG Antibodies against Three Antigens

To confirm the specificity of antibodies detected by the optimized ELISA, sera were pre-incubated in an excess of the selected mixture of antigens in soluble form and then applied to the “in-house” ELISA, as previously described. One pre-pandemic control who had tested positive in the developed ELISA, as well as two COVID-19 patients’ sera, were assessed. Samples diluted to a concentration (1:100) were pre-incubated with the selected three-antigen mixture in 10-times molar excess before being applied to the optimized experimental procedure.

### 2.9. Comparing the “In-House” SARS-CoV-2 ELISA to Commercially Available Assays

Patients’ sera were also evaluated against commercially available ELISA diagnostic kits with pre-coated ELISA plates. In detail, all COVID-19-positive patient samples (n = 44) and a pool of 150 COVID-19-negative patient samples were evaluated using the commercially available EDI, VIRCELL, NOVALISA and WANTAI serologic assays. EDI detects IgG antibodies, VIRCELL detects IgG and IgM/IgA combined antibodies, NOVALISA detects IgG, IgM and IgA antibodies separately and WANTAI detects combined IgG, IgM and IgA antibodies.

### 2.10. Implementation of the Developed ELISA in PCR-Positive Children

The optimized ELISA was also applied in sera from children aged 0–14 years of age with PCR-confirmed COVID-19 1–5 and 8–14 days after disease onset.

### 2.11. Statistical Analysis

Descriptive statistical analyses were performed to describe the IgG, IgA and IgM binding to different protein antigens, as recorded by the mean optical density across antigen replicates. All statistical analyses were performed using GraphPad Prism (version 7.0, GraphPad Software).

## 3. Results

### 3.1. ELISA Development

The most potent protein antigens were identified using an indirect ELISA among patients and controls, where recombinant SARS-COV-2 proteins were used as capture antigens (Figure 1). We found that RBD at 2.5 μg/mL, S1 at 1 μg/mL, S1S2 at 1 μg/mL, S2 at 1 μg/mL, M at 2 μg/mL and N at 1.5 μg/mL were the optimal concentrations because they yielded the greatest separation between OD values of COVID-19-positive and pre-COVID-19 control results using preliminary experiments. Notably, no immunoreactivity against the E antigen was shown, even when high concentrations (5 μg/mL) or different dilution buffers (PBS and carbonate–bicarbonate) were applied. Similarly, we determined the 1:100 concentration as the optimal serum dilution in PBS containing 2% BSA, because it sufficiently differentiated patients from controls.

Patients’ sera preferentially reacted with four protein antigen RBDs, S1S2 as a whole protein and S2 fragment and N, ranging from 79.55% to 84.09% of the sera (Figure 2). Control sera displayed low absorbance levels with no distinct signal peaks among individual peptides. S1 and M were recognized by 54.55% and 61.36% of the sera, respectively, whereas remarkably, no patient sera recognized the E protein antigen (Figure 2). The pre-COVID-19 controls were used to establish the background reactivity to the protein antigens for IgG antibodies. The cut-off value for each antigen was determined using the mean optical density plus 2 SDs of normal controls.

Subsequently, mixtures of all potential combinations of protein antigens were prepared and evaluated in terms of their antigenicity among patients’ sera (Figure 3). The mixtures evaluated displayed various immunoreactivities among patients’ sera. The vast majority of recovered COVID-19-positive serum samples reacted strongly with the mixture containing RBD + S2 + N (Figure 3). Importantly, all sera collected 8–14 days from disease onset recognized the mixture of RBD + S2 + N (Figure 4). Pre-COVID-19 controls exhibited lower levels of immunoreactivity in the majority of cases (below the mean OD + 2SD cut-off) (Figure 4). The cut-off value for this assay was determined using the mean optical density plus 2 SDs of normal controls and was used to determine assay sensitivity and specificity. The sensitivity of the optimized assay was 92%, whereas its specificity was 97%. Notably, the antibody sensitivity increased to 100% after 7 days from disease onset.

### 3.2. Optimization of SARS-CoV-2 Protein Coating Concentration and Serum Dilution

IgG reactivity was tested using decreasing concentrations of the most potent combination against four COVID-19 patients’ sera in a 1:100 dilution. Four serial dilutions of the combination were evaluated (1:1, 1:2, 1:4, 1:8). As shown in Figure 5, the highest OD obtained was at a 1:1 dilution. The samples taken from the pre-pandemic controls showed sustained baseline levels of IgG.

Next, the differences in the reactivity of the optimized ELISA of SARS-CoV-2 from RT-PCR-confirmed COVID-19 patients’ sera were assessed using two serial dilutions of 1:100 and 1:200 sera. Sera reactivity significantly decreased using the 1:200 concentration.

### 3.3. IgA and IgM Antibody Detection

The selected ELISA was used for the detection of IgA and IgM antibodies in COVID-19-positive patient samples as well as in pre-COVID-19 controls (Figure 6)**.** The detailed assay procedure was as described for IgG antibody detection. Notably, goat anti-human alkaline phosphatase-conjugated IgA antibody (Jackson ImmunoResearch Laboratories) was added at a concentration of 1:1500, instead of 1:3000, following preliminary experiments. The cut-off values for both IgM and IgA assays were determined using the mean optical density plus 2 SDs of normal controls. The IgM assay displayed sensitivity at 74%, whereas its specificity was 95%. The IgA assay displayed lower sensitivity and specificity, 70% and 91%, respectively. The IgM and IgA antibody positivity increased to 88% and 81%, respectively, after 7 days from symptoms onset.

### 3.4. Determining Inter-Assay and Intra-Assay Variability and Repeatability

Six COVID-19-positive patients’ samples (n = 6) and six pre-COVID-19 controls’ sera (n = 6) were tested four times individually to define the reproducibility of the optimized SARS-CoV-2 ELISA. Minimal acceptable inter-assay variability was found through the repetitive testing of IgG: 8.2% and 11.2%, on average, in patients’ and pre-COVID-19 controls’ sera, respectively. This trend was similar for IgA (11.6% and 8.2%), and IgM (11.2% and 8.1%). The precision of the assay was determined by performing intra-assay variability tests. Sera of six patients and six pre-pandemic controls were analyzed in eight replicates on the same plate using the “in-house” ELISA. The coefficient of variation (%CV, defined as the ratio between the standard deviation and mean value) was lower than 20% for all sera tested.

### 3.5. Inhibition of IgG Antibody Binding

To exclude the possibility that the antibody binding observed in the developed assay was caused by non-specific reactions, the reactivity of four COVID-19 patients’ sera (n = 4) and one antibody-positive sample from pre-COVID-19 controls (n = 1) for IgG was inhibited using the three-mixture antigens (RBD, S2, N) in excess solution. COVID-19-positive patient antibody binding was inhibited by 79% to 91% on average using excess antigens. One pre-COVID-19 control who tested positive was also inhibited to a similar degree (82%) as the COVID-19-positive samples.

### 3.6. Comparing the In-House SARS-CoV-2 ELISA to Commercially Available Assays

Patients’ sera were also evaluated in terms of their antigenicity using different commercially available serological diagnostic kits for IgG, IgA and IgM antibodies (Figure 7).

### 3.7. Evaluation of the “In-House” ELISA in Children

The “in-house” ELISA was evaluated in a pediatric population (n = 15, aged 0–14 years) with PCR-confirmed COVID-19, 8–14 days from disease onset and 30 age-matched controls (Figure 8). The sensitivity for IgG antibodies was 94%, whereas its specificity was 100%. The sensitivity and specificity for IgM and IgA antibodies were 87% and 96% and 61% and 85%, respectively. Notably, seven pediatric sera collected 1–5 days after disease onset were also evaluated; however, only one had detectable IgA, IgM and IgG antibodies.

## 4. Discussion

In this study, we describe a high-throughput, reproducible, easy-to-perform serological method to detect antibodies against SARS-CoV-2 using the immunogenic mixture of RBD, S2 and N proteins of the virus as a capture antigen. The above antigens were rationally selected through preliminary experiments, where different individual protein antigens and all potential combinations were used as capture antigens. RBD, S2 and N protein antigens were found to be the most immunogenic among patients’ sera, whereas their combination as capture antigens was recognized by the majority of patients’ sera.

Previous studies on other common human coronaviruses and extremely accurate SARS-CoV-2 genome sequencing at the very beginning of the pandemic determined the S [23] and the N [24] structural proteins as major targets of antibodies. Hence, the latter antigens are most commonly used as capture antigens by the currently available immunoassays for COVID-19 diagnosis and the evaluation of immune responses. The surface glycoprotein S, containing the RBD region, plays a key role in immunity [25,26] and has widely been used for vaccine development [27]. The RBD fragment (~200 amino acids) is located within the S1 fragment (~700 amino acids). In our study, the RBD was significantly more immunogenic compared to S1, because several epitopes may be buried within the core of a folded protein or at the binding interface of a ligand–receptor complex. The N protein is smaller than S, is without a glycosylation site, and is a leading antigen used in serological kits for serodiagnosis [28,29,30,31], due to its excessive expression in the course of infection [3,32,33] and early immune response [34,35]; however, its immunological importance is less well-documented. Several factors have been shown to affect the pattern of kinetics of antibody responses to S and N proteins (i.e., disease severity and age) [36,37]. The S2 subunit within the S protein mediates fusion between viral and host membranes with high amino acid sequence identity among different strains [38,39]. Thus, the fusion mechanism during virus infection is well-conserved, explaining the high level of immunogenicity of the S2 region among different patients’ sera in our study. The M protein is responsible for the overall shape of the viral envelope [40]. Only half of our patients exhibited an M protein. This is in accordance with previous studies showing that antibodies against the M protein are present in patients’ sera, albeit in lower levels compared to antibodies against S and N proteins, mainly rising at times later than 21 days post-infection [41,42]. The M protein may also harbor B cell epitopes that can mount a neutralizing antibody response, as suggested for SARS-CoV-1 [43,44]. The E protein is a small, integral membrane protein involved in several aspects of the viral life cycle, such as assembly, budding, envelope formation, and pathogenesis [17,45]. In our study, no antibodies were detected against the E protein. This finding is in line with previous studies where antibodies against E protein were rarely detected [46].

Early and accurate COVID-19 diagnosis is fundamental in contact tracing and epidemiology surveillance. Clinical manifestations play a key role in the early detection of COVID-19 cases. However, although COVID-19 can be a severe infectious disease, many patients will present few or no symptoms [4]. Currently, molecular techniques are the gold standard for accurate diagnosis. Nonetheless, serological assays, in contrast to molecular assay techniques, may even be useful for seroepidemiology purposes in asymptomatic individuals as well as vaccine immune response assessments [13]. Currently available commercial SARS-CoV-2 serological assays mainly detect antibodies against S and N protein antigens, with varying sensitivity and specificity, depending on the study population and time of performance. The IgM and IgG or combined IgG and IgM antibodies are the most widely used biomarkers for the detection of SARS-COV-2 infection in commercial ELISAs, which usually provide pre-coated plates. A recent metanalysis has reported that the sensitivities and specificities were generally high, ranging from 80% to 100% and 95% to 100% in most studies [47]. For all the IgG-based ELISA tests, the sensitivity estimates ranged from 65% to 100%, and specificity estimates ranged from 86% to 100%. In the IgM-based tests, the sensitivity and specificity in the individual studies ranged from 44% to 100% and 69% to 100%, respectively. Thus, the “in-house” ELISA using the mixture of the three SARS-CoV-2 antigens (RBD, S2 and N) as capture antigens displayed comparable and even higher sensitivity and specificity than commercially available ELISA diagnostic kits. Our “in-house” ELISA corresponds with other commercial assays that report high sensitivity for IgG antibodies, whereas lower levels of sensitivity were shown for IgA and IgM.

Assay specificity is also an important issue to be addressed. The cross-reactivity of antibodies with multiple coronaviruses needs to be considered when developing a SARS-CoV-2 serological ELISA [48,49,50,51]. Several currently available kits for the diagnosis of SARS-CoV-2 ELISAs have revealed cross-reactivity to other seasonal coronaviruses, including HCoV-OC43, HCoV-NL63, HCoV-229E and HCoV-HKU1, adversely affecting the specificity of the assays [24,25,26,27,52]. In addition, other features related to the host, including the existence of rheumatoid factor and heterophile antibodies, are likely to cross-react with SARS-CoV-2 antigens and generate false-positive results [53]. Misdiagnoses of common cold coronaviruses or other medical condition as SARS-CoV-2 may lead to overloaded hospitals while increasing the possibility of real infection by SARS-CoV-2 during otherwise preventable hospital admissions. In our assay, using the mixture RBD, S2 and N, only 4 out of the 150 pre-COVID 19 sera were screened as positive. We used a biobank containing pre-COVID-19 controls (n = 150) to determine the cut-off value of the optimized ELISA for IgG, IgA and IgM antibodies. The optimized ELISA described here has a specificity comparable to the specificity of the commercial ELISA evaluated in this study population.

Early serological assessments have shed light on antibody serology to COVID-19 proteins. Following exposure, IgG, IgA and IgM increase progressively; IgA and IgM levels peak at 7–20 days after disease onset, followed by IgG [54,55]. Assessing all three antibody isotypes, IgG, IgA, and IgM antibodies, immune responses induced at different stages were recorded, minimizing false negative results. Furthermore, upon infection, SARS-CoV-2 elicits distinct humoral responses against different protein antigens, RBD, S1 and S2 domains of the S glycoprotein and N protein [56,57,58]. However, the kinetics of antibodies against these proteins is not yet fully elucidated. Thus, the ELISA described here that combines different protein antigens as capture antigens may detect a higher proportion of infected individuals. In addition, our “in-house” ELISA is easy to perform, fast, does not need specific and expensive technical facilities and can be conducted in standard diagnostic laboratories.

Children are less susceptible to SARS-CoV-2, presenting with no or milder symptoms [59,60]. Infants are generally at a higher risk of more severe disease [61,62,63]. Children with underlying medical conditions, i.e., diabetes, asthma, or severe immunosuppression, may present with more severe symptoms [62,63,64]. In children, the incubation time seems to be 1.5–2 days longer than in adults [65]. Finally, mortality is lower (<0.1%) than that of adults (5–15%) [66]. Previous studies have shown that antibody levels among children are lower compared to adults, after disease onset. This could confirm the lower vulnerability of younger subjects to SARS-CoV-2 infection [61,67,68,69,70]. We found that children rarely have detectable antibodies 0–7 days after disease onset using the optimized ELISA, whereas the vast majority of infected children were screened as positive 8–14 days after disease onset. Thus, the optimized ELISA maintained its high sensitivity when applied in a pediatric population.

Our study has some limitations. One limitation of the study is that sera from patients positive for common human coronavirus strains, or other common respiratory viruses, were not evaluated. However, a large number of pre-pandemic controls were used to reduce the possibility of any cross-reactivity with SARS-CoV-2. In addition, it has been shown that other factors may affect assay specificity, i.e., the existence of rheumatoid factor and heterophile antibodies. Although such information was missing from our control group, we assume that it may only have had a slight effect on assay specificity, because only four pre-pandemic sera samples tested positive using the “in-house” ELISA.

In conclusion, the “in-house” ELISA that uses the mixture of the three antigens (RBD, S2 and N) as capture antigens displayed high sensitivity and specificity for IgG antibodies, and comparable sensitivity and specificity for IgM and IgA antibodies to other commercially available serological assays in this study population. In addition, the pattern of kinetics of immune response against different proteins is yet not well-defined, although may be affected by several factors (host individuality, disease severity, age and gender). Therefore, evaluating patients’ sera immunoreactivity against a combination of different antigens reduces the possibility of a false negative result, enhancing the sensitivity of the developed “in-house” assay. Considering its low cost, high reproducibility and ease-of-performance, this assay will complement RT-PCR tests and help in accurate and prompt disease diagnoses. The assay may support SARS-CoV-2 seroprevalence surveillance, which, in turn, is required for disease prevention and control at the population level.

## Figures and Tables

**Figure 1 diagnostics-11-01970-f001:**
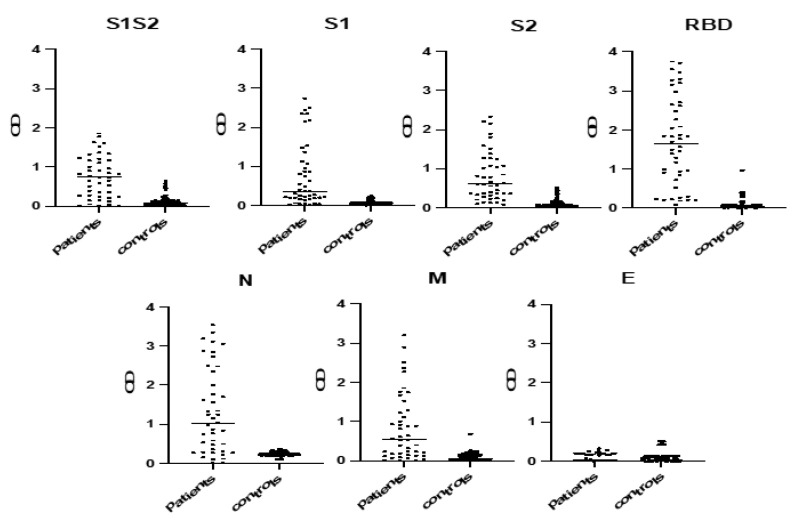
Immunoreactivity of different protein antigens in patients’ and controls’ sera. Sera (at a dilution 1:100) were tested against different capture antigens (RBD 2.5 μg/mL, S1 1 μg/mL, S1S2 1 μg/mL, S2 1 μg/mL, M 2 μg/mL, N 1.5 μg/mL and E 5 μg/mL). The cut-off value for each antigen was determined using the mean optical density plus 2 SDs of normal controls. Black symbols represent the Optical Density (OD) value of each serum at 405 nm.

**Figure 2 diagnostics-11-01970-f002:**
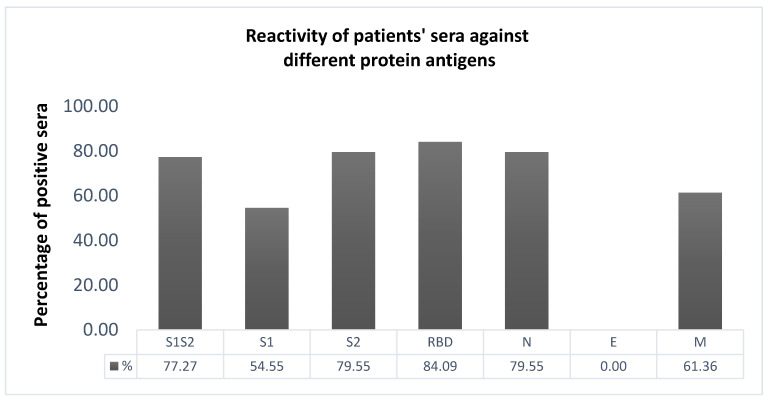
Prevalence of antibodies against different protein antigens in patients’ sera. The pre-COVID-19 controls were used to establish the background reactivity to the protein antigens for IgG antibodies. The cut-off value for each antigen was determined using the mean optical density plus 2 SDs of normal controls. Sera that yielded an OD value above that cut-off was considered as positive.

**Figure 3 diagnostics-11-01970-f003:**
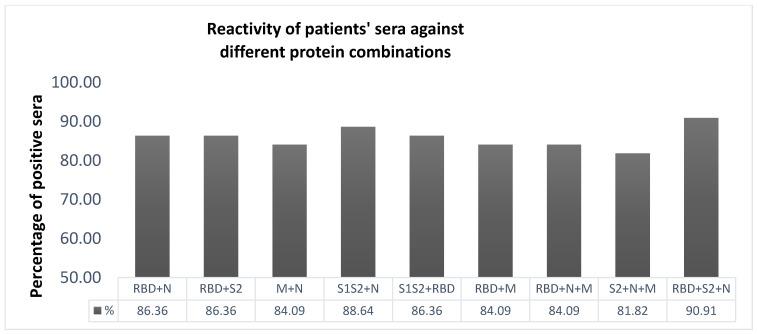
Prevalence of antibodies against different combinations of protein antigens in patients’ sera. Mixtures of all potential combinations of protein antigens were evaluated in terms of their antigenicity among patients’ sera (at a dilution of 1:100). The concentrations of antigens in every mixture were the same as when used separately (RBD 2.5 μg/mL, S1 1 μg/mL, S1S2 1 μg/mL, S2 1 μg/mL, M 2 μg/mL, N 1.5 μg/mL). The cut-off value for each antigen was determined using the mean optical density plus 2 SDs of controls.

**Figure 4 diagnostics-11-01970-f004:**
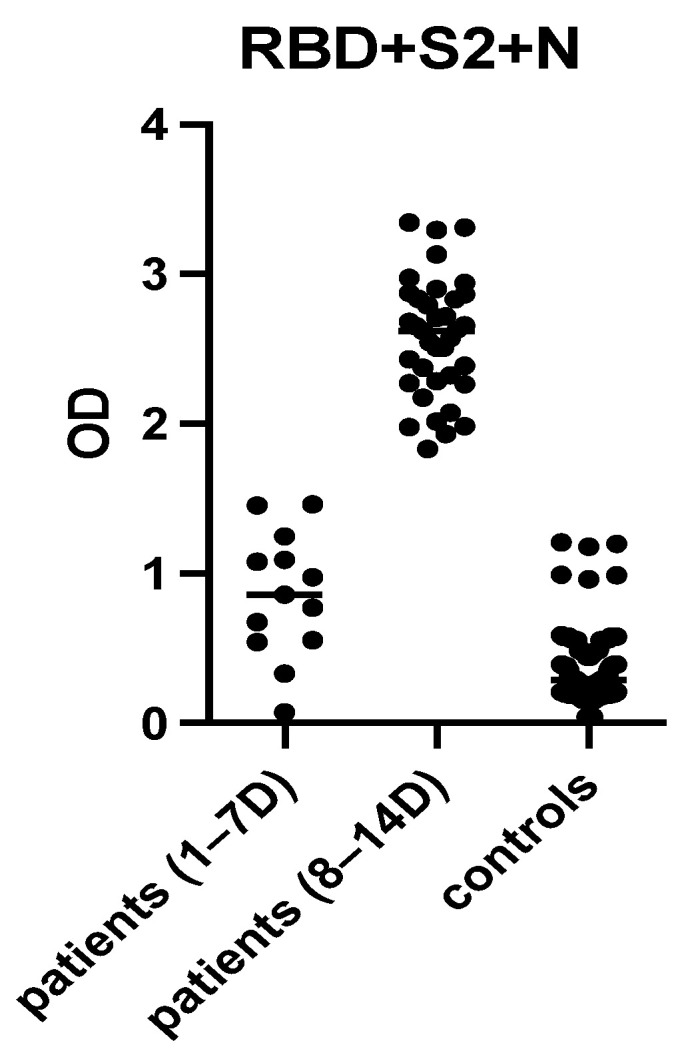
Immunoreactivity of the most potent combination of protein antigens among patients’ and controls’ sera (at a dilution of 1:100) using the “in-house” ELISA, where the combination of 3 different proteins (RBD 2.5 μg/mL, S2 1 μg/mL, N 1.5 μg/mL) was used as the capture antigen. Patients’ sera were divided in two groups obtained 1–7 days and 8–14 days after disease onset. The cut-off value for each antigen was determined using the mean optical density plus 2 SDs of controls. Black symbols represent the Optical Density (OD) value of each serum at 405 nm.

**Figure 5 diagnostics-11-01970-f005:**
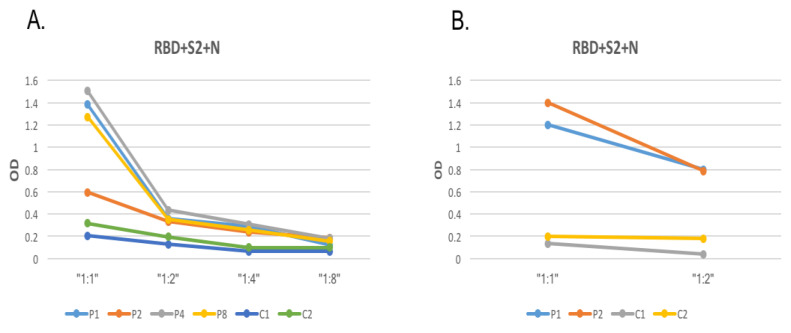
Optimization of SARS-CoV-2 protein coating concentration and serum dilution. (**A**) Four serial capture antigen dilutions (1:1, 1:2, 1:4, 1:8) were tested. (**B**) Two serial serum dilutions (1:100 and 1:200) were tested.

**Figure 6 diagnostics-11-01970-f006:**
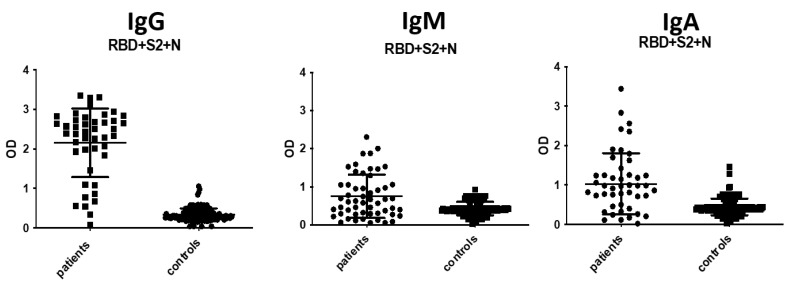
Immunoreactivity of patients’ and controls’ sera against RBD + S2 + N combination for the three different antibody isotypes. Sera were used at a dilution of 1:100, whereas the combination of the 3 different proteins contained RBD, S2 and N proteins at concentrations of 2.5 μg/mL, 1 μg/mL and N 1.5 μg/mL, respectively. The cut-off value for each group was determined using the mean optical density plus 2 SDs of controls. Black symbols represent the Optical Density (OD) value of each serum at 405 nm.

**Figure 7 diagnostics-11-01970-f007:**
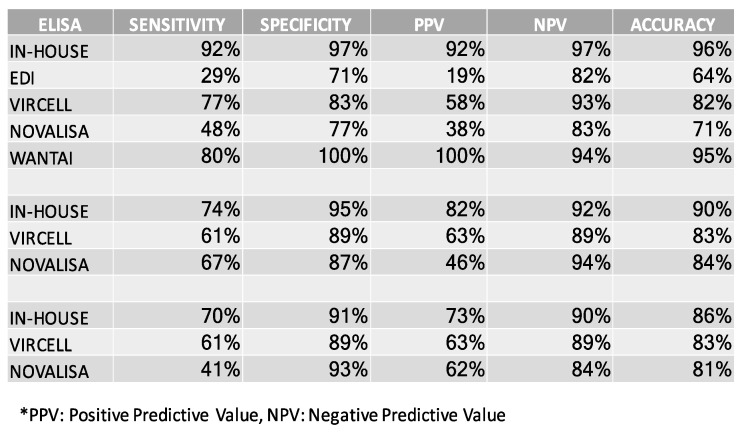
Tests characteristics of “in-house” ELISA and commercial ELISA kits. Three different antibody isotypes were evaluated.

**Figure 8 diagnostics-11-01970-f008:**
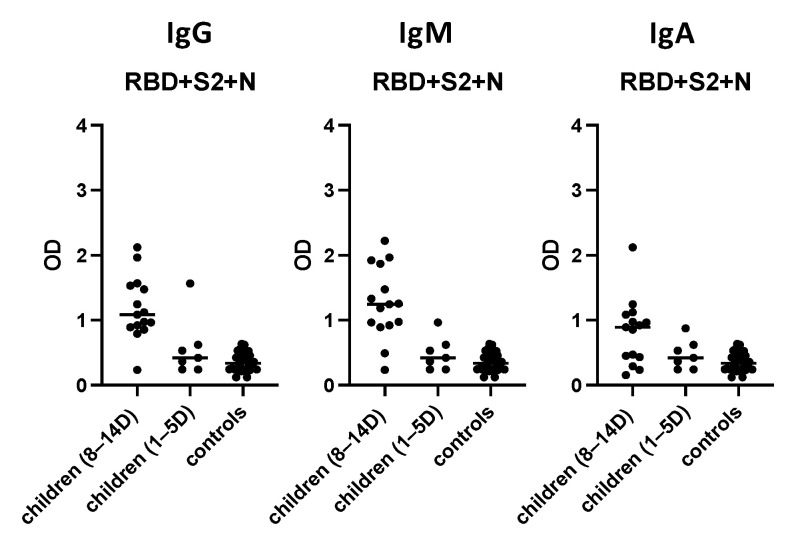
Immunoreactivity of sera from children with PCR-confirmed COVID-19 and controls. Sera were used at a 1:100 dilution using the “in-house” ELISA where the combination of 3 different proteins (RBD 2.5 μg/mL, S2 1 μg/mL, N 1.5 μg/mL) was used as the capture antigen. Black symbols represent the Optical Density (OD) value of each serum at 405 nm.

## Data Availability

Not applicable.

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
