# Peer review of "Development of an Enzyme-Linked Immunosorbent Assay (ELISA) for Accurate and Prompt Coronavirus Disease 2019 (COVID-19) Diagnosis Using the Rational Selection of Serological Biomarkers"

_diagnostics, 2021, doi:10.3390/diagnostics11111970_

Round 1

Reviewer 1 Report

Lagousi et al. aimed to develop a serological ELISA assay to measure antibodies against SARS-CoV-2 by combining different antigens. It’s very interesting study and useful for the society. Authors used enough number of Covid-19 patients’ sera and tested against commercial ELISA diagnostic kits. The In house, ELISA assay showed better sensitivity and specificity for the antigens. Overall, it’s a good progress, acceptable for publication, but figures 2 and 3 need to improve and remove the inner lines.  

Author Response

Dear reviewer,

Thank you for your comments.

Figures 2 and 3 have been changed accordingly. 

Reviewer 2 Report

Dear Editor,

thank you for the opportunity to review the manuscript. This is very interesting paper dealing with the development of new ELISA for detecting specific antibodies against new coronavirus. Since there are several coronaviruses circulating in humans it is very important to have the test highly specific. Therefore, although I think the manuscript is well written there are several points that need clarification.

First, in Materials and methods Study population is presented in the first (2.1) paragraph with Ethics approval without mentioning any sera from the children and then later in paragraph 2.10 implementation of the ELISA in PCR positive children is described. Please clarify this study group.

Results are nicely presented. However, the sera from known common coronavirus proved patients are missing. This would be very important to know the cross reactivity. Nevertheless, it can be stressed out in the discussion as the limitation of this study.

In the second paragraph of the Discussion section there is the sentence that starts with In the IgM based tests... and then the precentage range is presented as 695 to 100%. Please correct the number in the range.

It would also be interesting to know the affect od rheumatoid factor and heterophile antibodies on the ELISA results or to emphasize the limitation of the study.

References list is without the numbers and in the text references are cited as numbers. Please check the journal requirements for the citation of the literature.

Best regards!

Reviewer 3 Report

Dear authors,

Nice work

Here my inputs step-to-step.

M&M

Study population

In this study the n=44, could be possible to include more samples?, To get 100 samples will be good to get a strong stsistical analyses. I can understand that maybe It is too hard to get thenm however perhaps 50 ¿?

Proteins

Do you know the author the purity grade of each protein? Could it be specified ?

Determining inter-assay variability and repeatability

Why only n=2 ?, What has happened with the intravariabililty?

Implementation of the developed ELISA assay in PCR-positive children

How many samples ? As for this chapter I suggest to show the results apart.

Results

I would apreciate some specific explanation for each figure and not only a short headline, and importatly the same scale for each group of figure, please. Improve figure 4 and expamde the explanation, like for the others

For example, figure 1

explain and specificy the concentrations used,  the cut off to be positive, not only the mean!

DISCUSSION: I have missed a deep discussion e.g., why the E protein does not shown reactivity. I would apreciate a real discussion about the differences and different proteins and the reasons about the different immunogenicty rather than a soft decriptive ELISA assay. SO,please, imporved this chapter, why I should use this assay to diagnos COVID-19.

Thanks

Round 2

Reviewer 2 Report

Dear Editor,

Authors have accepted all comments and updated the manuscript accordingly. If some data were missing, the limitations of the study are transparently descibed in the discussion.

Therefore, the methods, results and discussion are improved and data beter presented.

I would just suggest authors to read once more the clean text after acception of changes to check the possible typing errors.

Author Response

Thank you for your comment.

The final text has been assessed t check for any typing errors.

Reviewer 3 Report

Please, reiew figure 4. the quality is not too good.

Thanks

Author Response

Thank you for your comment. A title has been added on the top of the figure representing the mixture of antigens used. The size of the symbols have been increased. The space between columns has been increased as well. Finally the scatter plot appearance has been changed to minimize overlap of different data points. The size of the figure has been changed to limit figure deformations.